# Dissecting the roles of calcium cycling and its coupling with voltage in the genesis of early afterdepolarizations in cardiac myocyte models

Rui Wang[1], Zhilin Qu[2], Xiaodong Huang[1] *

**1** Department of Physics, South China University of Technology, Guangzhou, China, **2** Department of Medicine, David Geffen School of Medicine, University of California, Los Angeles, Los Angeles, California, United States of America

* schuangxd@scut.edu.cn

## Abstract

Early afterdepolarizations (EADs) are abnormal depolarizations during the plateau phase of the action potential, which are known to be associated with lethal arrhythmias in the heart. There are two major hypotheses for EAD genesis based on experimental observations, i.e., the voltage ($V_m$)-driven and intracellular calcium (Ca)-driven mechanisms. In ventricular myocytes, Ca and $V_m$ are bidirectionally coupled, which can affect each other's dynamics and result in new dynamics, however, the roles of Ca cycling and its coupling with $V_m$ in the genesis of EADs have not been well understood. In this study, we use an action potential model that is capable of independent $V_m$ and Ca oscillations to investigate the roles of $V_m$ and Ca coupling in EAD genesis. Four different mechanisms of EADs are identified, which are either driven by $V_m$ oscillations or Ca oscillations alone, or oscillations caused by their interactions. We also use 5 other ventricular action potential models to assess these EAD mechanisms and show that EADs in these models are mainly $V_m$-driven. These mechanistic insights from our simulations provide a theoretical base for understanding experimentally observed EADs and EAD-related arrhythmogenesis.

## Author summary

Early afterdepolarizations (EADs) are dangerous abnormal electrical activities in the heart, which may cause lethal arrhythmias. Although EADs have been widely investigated in both experimental and computer simulation studies, their mechanisms remain incompletely understood. In the present work, we carry out computer simulations using action potential models with detailed formulations of ionic currents and calcium cycling to investigate the genesis of EADs. Different mechanisms and causes are identified in our simulations, which agree with experimental observations. The mechanistic insights from our work provide a theoretical base for understanding the mechanisms of EADs and EAD-related arrhythmogenesis.

**Funding:** This work is supported by the Guangdong Basic and Applied Basic Research Foundation under Grant No. 2021A1515010500 (XH); National Institutes of Health grants R01 HL134709, R01 HL139829, R01 HL157116, and P01 HL164311 (ZQ). The funders had no role in study design, data collection and analysis, decision to publish, or preparation of the manuscript.

**Competing interests:** The authors have declared that no competing interests exist.

## Introduction

Early afterdepolarizations (EADs) are secondary transmembrane voltage ($V_m$) depolarizations of a cardiac myocyte during the plateau and repolarizing phases [1, 2]. EADs are associated with lethal arrhythmias under certain diseased conditions, such as Torsade de Pointes in long QT syndrome (LQTS) [3–7]. Due to its importance in cardiac arrhythmogenesis, the causes and mechanisms of EADs have been comprehensively investigated in experiments [8–13], computer simulations [14–21], and bifurcation theories [22–27].

Hitherto there are two dominant hypotheses of EAD generation mechanisms [2, 28, 29]. The first one is the voltage-driven ($V_m$-driven) mechanism in which EADs are secondary depolarizations driven by instabilities in the voltage system. In this mechanism, the induction of EADs is attributed to two major factors, i.e., reduced repolarization reserve (RRR) and reactivation of L-type calcium (Ca) current ($I_{Ca,L}$) [5, 9]. RRR is a condition in which the outward currents are reduced and/or the inward currents are increased. For example, in LQTS type 1 and 2, potassium currents are reduced, while in LQTS type 3, the late component of the sodium current is increased. RRR decelerates repolarization and thereby provides enough time for reactivation of $I_{Ca,L}$ to result in depolarizations in the repolarization phase. Based on nonlinear dynamics analyses, RRR combining proper $I_{Ca,L}$ kinetics can give rise to a Hopf bifurcation to lead to limit cycle oscillations in the plateau or repolarizing phase of the action potential (AP), manifesting as EADs [22, 24].

The second one is the Ca-driven mechanism in which instabilities in the intracellular Ca cycling system give rise to Ca oscillations or spontaneous Ca releases to generate EADs. Evidence of this mechanism has been demonstrated in several experimental studies [29–32]. The physiological hypothesis for this mechanism is as follows: Spontaneous Ca releases elevate the intracellular Ca level which increases the sodium-calcium-exchange (NCX) current ($I_{NCX}$) to depolarize $V_m$, manifesting as EADs. In this case, Ca oscillations are the primary cause and $V_m$ oscillations are a passive result of Ca oscillations. Simulation studies (e.g., Ref. [33–35]) have also been carried out to investigate this mechanism.

Although rigorous bifurcation analyses [20, 22–27] have revealed the $V_m$-driven instabilities induced bifurcations leading to EADs, the dynamical mechanisms and bifurcations for Ca-driven EADs have not been well understood. Moreover, $V_m$ and Ca are coupled via Ca-dependent ionic currents, and the coupling can either potentiate or suppress each other's instabilities, or bring in completely new dynamics [36, 37]. Therefore, a more rigorous analysis of Ca-driven instabilities and the roles of the interaction of Ca and $V_m$ in the genesis of EADs are necessary. To investigate the roles of Ca-driven instabilities in EAD genesis, one needs an AP model capable of exhibiting spontaneous Ca oscillations. Most of the AP models do not exhibit this behavior, but a recent model developed by Wilson et al [34] (called WG model in the present paper) incorporates a Ca cycling model that exhibits spontaneous oscillations and was used to investigate the Ca-driven mechanism of EADs. In this study, we use the WG model to perform systematic analyses on the roles of Ca and $V_m$ coupling in the genesis of EADs. Our analysis reveals 4 mechanisms of EADs: 1) EADs driven by $V_m$ oscillations alone; 2) EADs driven by Ca oscillations alone; 3) EADs driven by a Ca-$I_{Ca,L}$-$V_m$-Ca feedback loop; and 4) EADs driven by a large Ca transient. We also carry out simulations of 5 other ventricular AP models and observe that the majority of EADs in these models are via the $V_m$-driven mechanism, only a very small portion are related to the Ca-driven mechanism.

## Models and methods

The WG model is a rabbit ventricular myocyte AP model modified from the Shannon and Bers (SB) model [38], which can be presented in the general formulation as follows:

$$
\begin{cases}
\dfrac{dV_m}{dt} = -I_{ion}(V_m, \mathbf{y}, \mathbf{C}) + I_{sti} \\[2mm]
\dfrac{d\mathbf{y}}{dt} = \dfrac{\mathbf{y} - \mathbf{y}_\infty}{\tau_y} \\[2mm]
\dfrac{d\mathbf{C}}{dt} = \mathbf{F}(\mathbf{I}_z, \mathbf{C}, \mathbf{P}) \\[2mm]
\dfrac{d\mathbf{P}}{dt} = \mathbf{H}(\mathbf{C}, \mathbf{P})
\end{cases}
\tag{1}
$$

where $V_m$ is the transmembrane voltage, and $\mathbf{y}$ is a vector representing the gating variables (i.e., $\mathbf{y} = \{m, h, d, f, \ldots\}$, and each one of them controls the activation or inactivation of the corresponding ion channel). $\mathbf{C}$ represents the set of concentrations of different ions in different compartments of the cell (i.e., $\mathbf{C} = \{[Ca^{2+}]_{JSR}, [K^+]_{JSR}, [Na^+]_{JSR}, [Ca^{2+}]_{JXN}, [K^+]_{JXN}, [Na^+]_{JXN}, \ldots\}$, and the compartments are schematically shown in Fig 1A). $\mathbf{I}_z$ is the set of transmembrane ionic currents which modulates the corresponding intracellular ion cyclings (i.e., $\mathbf{I}_z = \{I_{CaL,JXN}, I_{Ks,JXN}, I_{NCX,JXN}, I_{CaL,SL}, I_{Ks,SL}, I_{NCX,SL}, \ldots\}$). $\mathbf{P}$ is the set of opening probabilities of the ryanodine receptor (RyR) gates controlling Ca release from junctional sarcoplasmic reticulum (JSR). $I_{ion}$ is the sum of the transmembrane ionic currents, and $\mathbf{F}$ and $\mathbf{H}$ stand for the functions of the corresponding variables. A schematic diagram of the WG model is shown in Fig 1A. For detailed mathematical formulations and definitions of the variables, one is referred to the original paper by Wilson et al [34].

In order to investigate the roles of Ca cycling and its coupling with $V_m$ in the genesis of EADs, we modulate several major parameters associated with Ca cycling and $V_m$. Note that the model divides the submembrane space into two compartments, i.e., the junctional (JXN) space and subsarcolemmal (SL) space (as colored by grey and light grey in Fig 1A). The formulation and maximal conductance of each current in both compartments are identical, i.e., $P_{Ca,JXN} = P_{Ca,SL} = P_{Ca}$, $G_{Ks,JXN} = G_{Ks,SL} = G_{Ks}$ and $\bar{I}_{NCX,JXN} = \bar{I}_{NCX,SL} = \bar{I}_{NCX}$. The parameters we vary are: $k_{max}$ (the maximal SR release rate constant controlling releasing flux via RyR, control $k_{max}$ = 0.2 ms$^{-1}$), the diffusive strength of $Ca^{2+}$ between SL and cytoplasm, control $J_{Caslmyo}$ = 7.4485*10$^{-13}$ l/ms), $P_{Ca}$ (permeability to Ca which controls the intensity of $I_{Ca,L}$, control $P_{Ca}$=0.00027 cm/s), $G_{Ks}$ (maximal conductance of slowly inward rectified potassium channel, control $G_{Ks}$ = 1.23 mS/$\mu$F), and $\bar{I}_{NCX}$ (NCX current density, control $\bar{I}_{NCX}$ = 5 A/F). These parameters are denoted in Fig 1A by red, and their associated formulations are given in the original publications [34, 38]. The AP of the control model is shown in Fig 1B. An adjustable factor $\alpha(z)$ (z stands for any one of the adjustable parameters) is multiplied to the control value. The other parameters are set as default unless specified otherwise.

In this model, due to fast diffusion between the two submembrane compartments, $[Ca^{2+}]_{JXN}$ and $[Ca^{2+}]_{SL}$ keep nearly identical, and thus we just treat them as one variable as $[Ca]_{sub}$ (i.e., $[Ca^{2+}]_{JXN} = [Ca^{2+}]_{SL} = [Ca]_{sub}$). Clamp of $[Ca]_{sub}$ means simultaneous clamp of $[Ca^{2+}]_{JXN}$ and $[Ca^{2+}]_{SL}$. The explicit Euler method is used to integrate Eq 1 with a time step 0.01 ms. The stimulus current density $I_{sti}$ is a pulse of 1 ms duration and 40 pA/pF magnitude.

For simplicity, we use a single set of initial condition and apply a single stimulus at t = 0 for most of the simulations to assess the EAD behaviors. The initial condition is presented in Supplemental Information (S1 Text). We have also used other pacing protocols (i.e., S1S1 and S1S2 pacing) to assess the EAD behaviors. To dissect out the EAD mechanisms, we use voltage

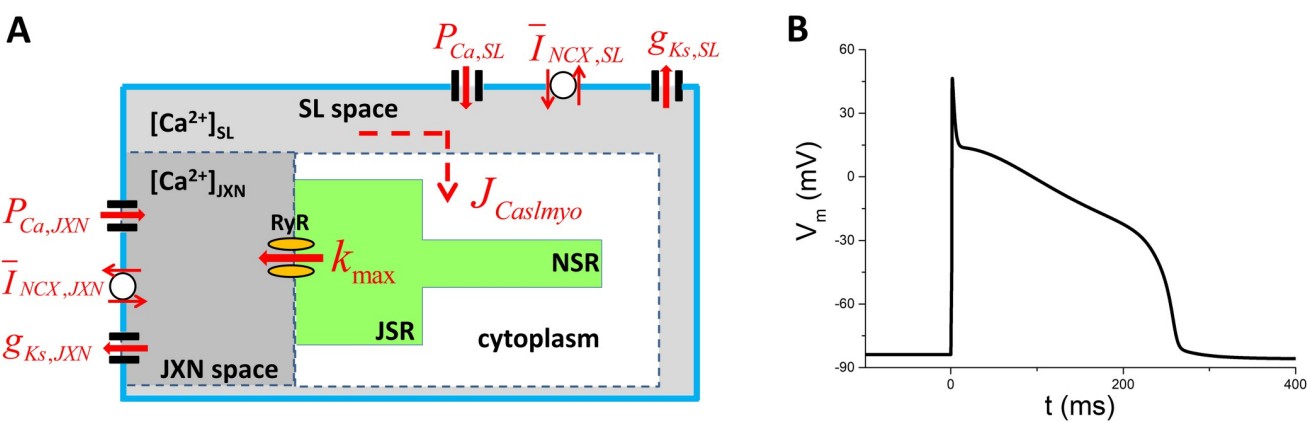

**Fig 1. The WG model.** A. A schematic plot of the WG model. The outer solid line represents the membrane. The space between the dashed line and the solid line is the submembrane space which is divided into two compartments: the junctional (JXN) and subsarcolemmal (SL) space. The JXN space (grey) is the cleft adjacent to the junctional sarcoplasmic reticulum (JSR, green), and the SL space (light grey) is the remaining submembrane space. NSR is the network sarcoplasmic reticulum. B. The AP of the control case.

clamp, Ca clamp, and ionic current clamp in a case-by-case manner, as will be shown in each EAD mechanism.

We also perform simulations using 5 other AP models. The control parameters and initial conditions for each model are presented in S1 Text.

## Results

### EADs driven by $V_m$ oscillations (type I)

We first set $k_{max}$ to be relatively small and $J_{Caslmyo}$ to be relatively large, which give rise to a low $[Ca]_{sub}$ due to an attenuated release and a rapid diffusion between the submembrane space and cytoplasm. Fig 2A is a phase diagram showing the behaviors in the $\alpha(P_{Ca}) - \alpha(G_{Ks})$ plane. The upper left grey region represents repolarization failure, and the lower right white region is normal repolarization. Three mechanisms of EADs are identified in this phase diagram, which are colored by dark yellow, green, and magenta. In the dark yellow region, the EADs are caused by the traditional $V_m$-driven oscillations. Fig 2B shows a representative AP. During the EAD phase, the $V_m$ turning point from the decreasing phase to the rising phase occurs earlier than that of $[Ca]_{sub}$, an indication that $V_m$ leads the oscillations, i.e., the oscillations are originated from instabilities in the $V_m$-system. Moreover, the rising of $V_m$ is with $I_{Ca,L}$ but not $I_{NCX}$, indicating that EADs are caused by reactivation of $I_{Ca,L}$. To further show that the oscillations are originated from $V_m$, we clamp $[Ca]_{sub}$ at different values within the oscillation range of $[Ca]_{sub}$ (grey region in Fig 2B) during the EAD phase to dissect out the role of Ca cycling (Fig 2C). After clamping $[Ca]_{sub}$, $V_m$ oscillations still occur although $V_m$ may either repolarize much later or fail to repolarize. These results indicate that under this parameter setting, the EADs are driven by oscillations from $V_m$, which is the well-known mechanism of EADs that has been investigated widely [5, 9, 20–22, 24].

In Fig 2A, the EAD behaviors are obtained from the same initial condition. It is known that the cardiac AP dynamics depends on initial conditions as well as pacing protocols [19, 21, 39, 40]. To show the results in Fig 2A are representative, we also use two other pacing protocols to assess the EAD behaviors. In the first protocol, the cell is paced with a fixed pacing period, called the S1S1 pacing protocol. Fig 3A is a phase diagram obtained using this pacing protocol with the S1S1 interval being 3 s. Fig 3C plots $V_m$ and $[Ca]_{sub}$ versus time showing an example

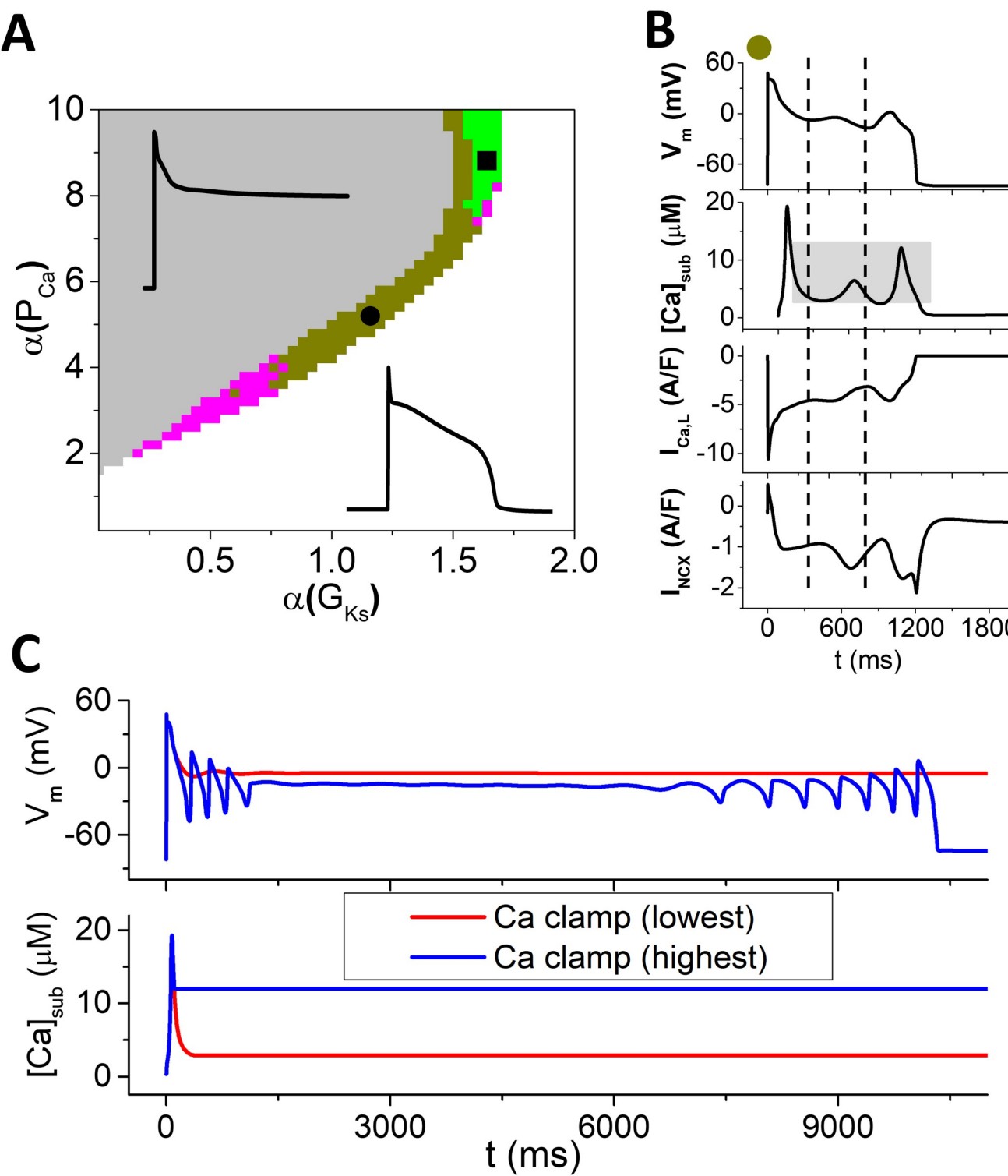

**Fig 2. EADs driven by $V_m$ oscillations.** A. Phase diagram showing different EAD behaviors in the $\alpha(P_{Ca}) - \alpha(G_{Ks})$ plane for $\alpha(k_{max}) = 2$, $\alpha(J_{Caslmyo}) = 2.64$, and $\alpha(\bar{I}_{NCX}) = 1$. The grey and white regions represent repolarization failure and normal repolarization, respectively. Insets show two representative APs. Three types of EADs are observed, as colored dark yellow, green, and magenta. B. A representative AP and associated currents in the dark yellow region (i.e., EADs driven by $V_m$ oscillations). The parameter set is $\alpha(G_{Ks}) = 1.16$ and $\alpha(P_{Ca}) = 5.4$, as marked by the black circle in A. The dashed vertical lines indicate the takeoff moments of the two EADs. The grey patch indicates the range of $[Ca]_{sub}$ clamped used in the $[Ca]_{sub}$-clamp simulations. C. An example showing $V_m$ and $[Ca]_{sub}$ versus time with $[Ca]_{sub}$ being clamped at the lowest (red) and highest (blue) levels.

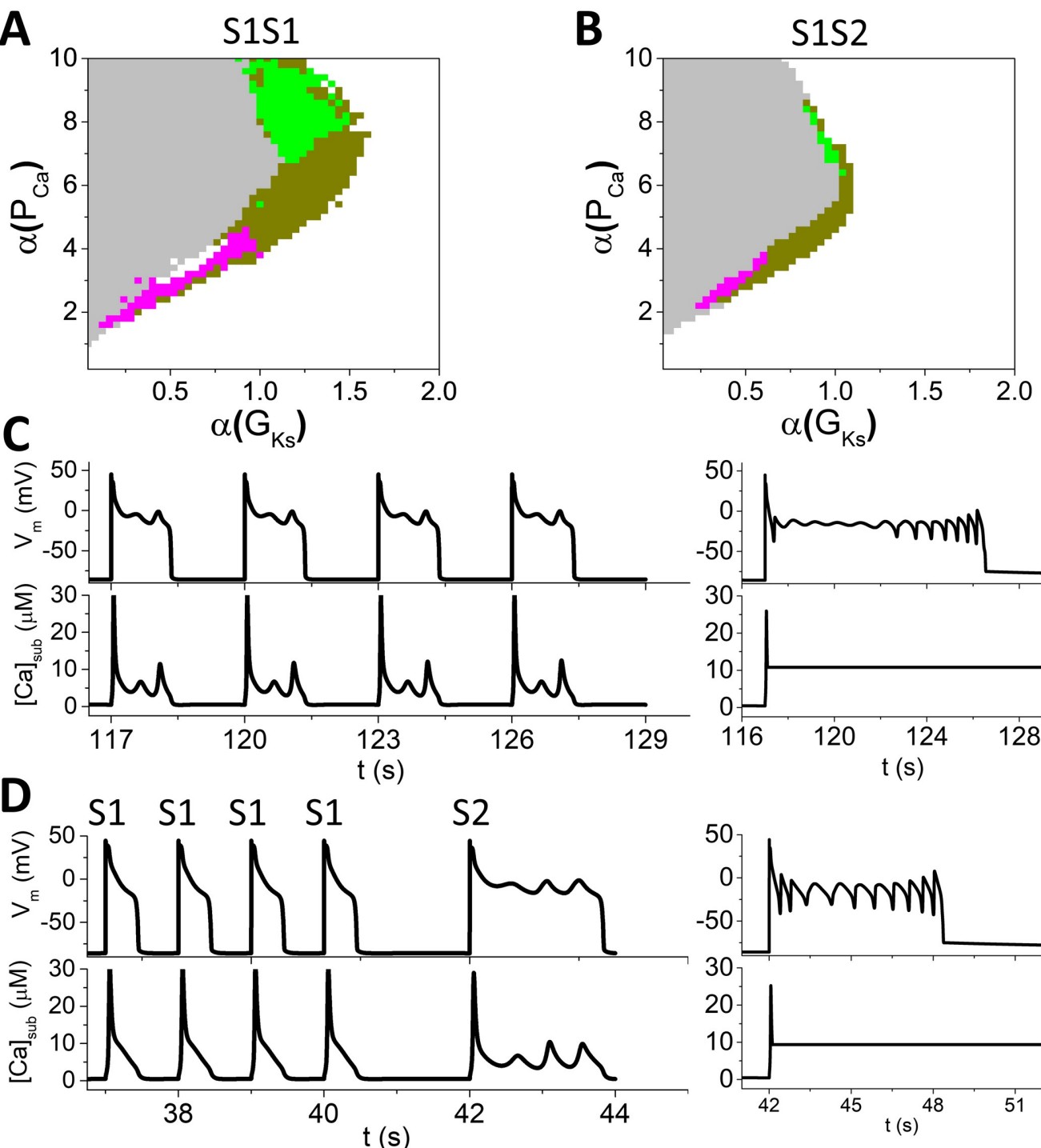

**Fig 3. EAD behaviors under S1S1 and S1S2 pacing protocols.** A. Phase diagram obtained with the S1S1 protocol. The parameter set is the same as Fig 2A. S1S1 interval = 3 s. The grey region is determined using the criterion of APD being greater than the S1S1 interval. The colored regions are different EAD regions classified by the same way as in Fig 2A. B. Phase diagram obtained with the S1S2 protocol. The EAD classification is done using the S2 beat. The S1S1 interval is 1 s and the S1S2 interval is 2 s. C. An example showing $V_m$ and $[Ca]_{sub}$ versus time from the regime of $V_m$-driven EADs with the S1S1 protocol. $\alpha(G_{Ks})$=0.8 and $\alpha(P_{Ca})$=4.4. The traces in the right panels are those when $[Ca]_{sub}$ is clamped in the last pacing beat. D. An example showing $V_m$ and $[Ca]_{sub}$ versus time from the regime of $V_m$-driven EADs with the S1S2 protocol. $\alpha(G_{Ks})$=0.8 and $\alpha(P_{Ca})$=4.6. The traces in the right panels are those when $[Ca]_{sub}$ is clamped in the S2 beat.

of $V_m$-driven EADs. In the second protocol, the cell is first paced with a fixed pacing period for many beats and then the pacing interval is lengthened for the last pacing beat, called the S1S2 pacing protocol. Fig 3B is a phase diagram obtained using this pacing protocol with the S1S1 interval being 1 s and the S1S2 interval being 2 s. Fig 3D plots $V_m$ and $[Ca]_{sub}$ versus time showing an example of $V_m$-driven EADs. Note that the grey regions in both cases are defined when the AP duration (APD) is longer than the S1S1 interval. This is because when APD is longer than the pacing period, it can be true repolarization failure, but it can also be 2:1 block and other complex behaviors (such as chaos) which complicate the identification of the EAD mechanisms. Nevertheless, although the parameters for the EAD behaviors are different, the structures of the phase diagrams obtained using these two pacing protocols are similar to the one in Fig 2A. This indicates that using the single initial condition with a single stimulus is an appropriate stimulation protocol for dissecting the EAD mechanisms. We will use this protocol for all simulations shown later in this study.

## EADs driven by Ca oscillations (type II)

In the magenta region in Fig 2A where $P_{Ca}$ is low, EADs are caused by Ca oscillations alone. Since the EAD amplitude in the parameter regime of Fig 2A is relatively small, we select another parameter set for $\alpha(k_{max})$, $\alpha(J_{Caslmyo})$, $\alpha(\bar{I}_{NCX})$ (all are increased) to better illustrate the properties of this type of EADs. Fig 4A is a phase diagram showing the EAD behaviors in the $\alpha(P_{Ca}) - \alpha(G_{Ks})$ plane. A representative case (star in Fig 4A) is shown in Fig 4B, where $V_m$, $[Ca]_{sub}$, and $I_{NCX}$ oscillate completely in-phase (aligned by the vertical dashed lines), whereas $I_{Ca,L}$ oscillates out-of-phase with them. This implies that the EADs are originated from the Ca oscillations not by reactivation of $I_{Ca,L}$ as in the EADs driven by $V_m$ oscillations shown in Fig 2. To further verify this, we carried out simulations under different clamp conditions. Fig 4C shows $V_m$ and $[Ca]_{sub}$ versus time for $[Ca]_{sub}$ is clamped during the EAD phase, and no $V_m$ oscillations could occur. On the other hand, if $V_m$ is clamped during the EAD phase (Fig 4D), $[Ca]_{sub}$ still oscillates. Moreover, if $I_{Ca,L}$ is clamped at a constant level (horizontal dashed line in Fig 4B), $[Ca]_{sub}$ and $V_m$ oscillations still maintain (upper panel in Fig 4E). If $I_{NCX}$ is clamped at a constant level (horizontal dashed line in Fig 4B), $[Ca]_{sub}$ still oscillates but not $V_m$ (lower panel in Fig 4E). Therefore, these results demonstrate that Ca oscillations are the primary driver of the EADs via $I_{NCX}$, agreeing with the experimental observations [32].

## EADs driven by the Ca-$I_{Ca,L}$-$V_m$-Ca feedback loop (type III)

When $P_{Ca}$ is large (the green region in Fig 2A), a new type of EAD mechanism is observed. Fig 5A is a representative case of this type of EADs (the square in Fig 2A). In this case, unlike the one driven by Ca oscillations, $V_m$ and $[Ca]_{sub}$ oscillations are out-of-phase. When either $[Ca]_{sub}$ (Fig 5B) or $V_m$ (Fig 5C) is clamped, neither $V_m$ nor Ca exhibits oscillations. This implies that neither the $V_m$ system alone nor the Ca system alone can oscillate, but coupling of $V_m$ and Ca promotes the instability for oscillations.

We hypothesize that the mechanism of this type of EADs is driven by a feedback loop between Ca and $V_m$ mediated by $I_{Ca,L}$. Specifically, elevation of $[Ca]_{sub}$ suppresses $I_{Ca,L}$ via Ca-dependent inactivation (via $f_{Ca}$ gate in the model). In the supplemental Fig A in S1 Text, we plot peak $I_{Ca,L}$ versus $[Ca]_{sub}$ under different holding $V_m$ for the WG model and other models, which shows that the $I_{Ca,L}$ amplitude decreases as $[Ca]_{sub}$ increases for holding $V_m < 40$ mV. Therefore, depending on the relative contributions of $I_{Ca,L}$ and $I_{NCX}$ ($I_{NCX}$ is overall smaller than $I_{Ca,L}$, compare Fig A with Fig D in S1 Text), elevation of $[Ca]_{sub}$ may affect $V_m$ negatively. Conversely, $V_m$ also affects $[Ca]_{sub}$ by modulating $I_{Ca,L}$ and $I_{NCX}$. In the WG model, increasing holding $V_m$ from -40 mV to +40 mV first decreases $[Ca]_{sub}$ ($V_m$ from -40 mV to -25 mV), then

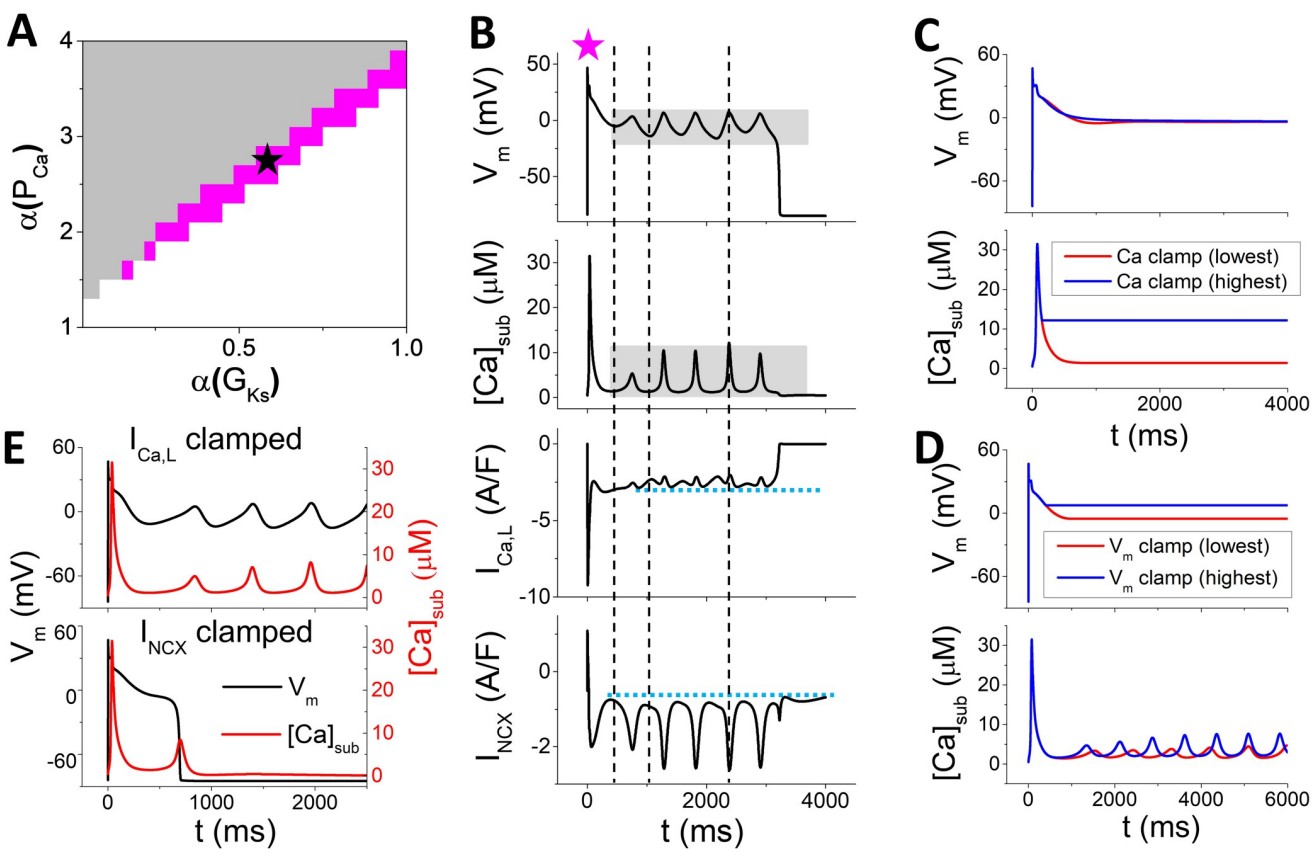

**Fig 4. EADs driven by Ca oscillations alone.** $\alpha(k_{max}) = 7$, $\alpha(J_{Caslmyo}) = 3.6$, and $\alpha(I_{NCX})=2.2$. A. Phase diagram in the $\alpha(P_{Ca}) - \alpha(G_{Ks})$ plane. EADs occur in the magenta region. B. An example showing $V_m$, $[Ca]_{sub}$, $I_{Ca,L}$, and $I_{NCX}$ versus time from the EAD region $[\alpha(G_{Ks}) = 0.56$, $\alpha(P_{Ca})=2.8$, marked as star in A]. The first two vertical dashed lines indicate the takeoff of the first two EADs and the third vertical dashed line marks the peak of the EAD prior to the last one. C. The same case as in B but $[Ca]_{sub}$ is clamped at two different levels in the plateau (grey patch in the $[Ca]_{sub}$ panel in B). D. The same case as in B but $V_m$ is clamped at two different levels in the plateau (grey patch in the $V_m$ panel in B). E. Upper panel: $V_m$ and $[Ca]_{sub}$ versus time when $I_{Ca,L}$ is clamped at a constant in the plateau (dashed line in the $I_{Ca,L}$ panel in B). Lower panel: $V_m$ and $[Ca]_{sub}$ versus time when $I_{NCX}$ is clamped at a constant in the plateau (dashed line in the $I_{NCX}$ panel in B).

increases $[Ca]_{sub}$ ($V_m$ from -25 mV to +10 mV) and decreases $[Ca]_{sub}$ after $V_m$ is above +10 mV (see the black dots in Fig 5D). This is caused by the effect of $V_m$ on $I_{Ca,L}$ in which the $I_{Ca,L}$ magnitude exhibits the same relationship with $V_m$ [see Fig B(f) in S1 Text]. This characteristic stems from the steady-state inactivation curve ($f_{ss}$) of $I_{Ca,L}$. These results show that in the $V_m$ range from -40 mV to +10 mV, $[Ca]_{sub}$ affects $I_{Ca,L}$ negatively (Fig A in S1 Text) and thus the $[Ca]_{sub}$ effects on $V_m$ are negative, but the $V_m$ effects on $[Ca]_{sub}$ can be either positive or negative (Fig 5D). Therefore, the feedback between $V_m$ and Ca in the $V_m$ range from -40 mV to +10 mV can be either positive or negative. Without a rigorous nonlinear dynamics analysis (the AP model is too complex to be useful for performing such an analysis), we cannot pinpoint the exact underlying mechanism. Nevertheless, one can postulate that these feedback loops may play important roles in the genesis of this type of EADs and understand this loosely as follows. In the AP plateau, a high $I_{Ca,L}$ maintains $V_m$ in the plateau. Elevation of $[Ca]_{sub}$ suppresses $I_{Ca,L}$ so that $V_m$ repolarizes partially (see the left vertical line in Fig 5A which aligns with the peak of $[Ca]_{sub}$). Then $[Ca]_{sub}$ falls back in response to $V_m$ repolarization. Lower $V_m$ can cause $I_{Ca,L}$ reactivation, and thereby depolarizes $V_m$ back and elevates $[Ca]_{sub}$ again (see the right vertical line in Fig 5A which aligns with the valley of $[Ca]_{sub}$). This chain process

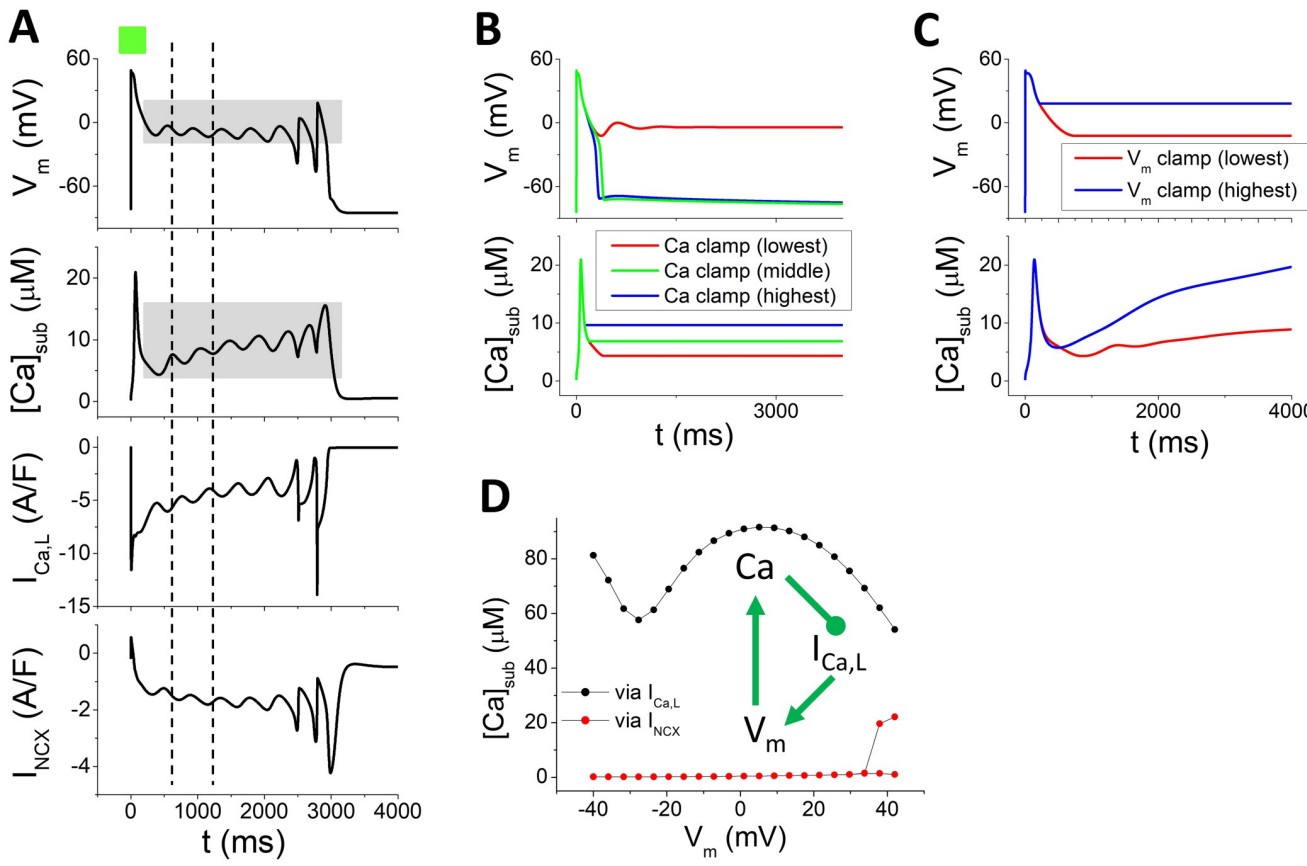

**Fig 5. EADs driven by oscillations caused by the Ca-$I_{Ca,L}$-$V_m$-Ca feedback loop.** A. $V_m$, $[Ca]_{sub}$, $I_{Ca,L}$, and $I_{NCX}$ versus time for a representative case [$\alpha$ ($G_{Ks}$) = 1.6 and $\alpha(P_{Ca})$ = 8.2, square in Fig 2A]. The left vertical dashed line indicates the peak and the right one indicates the valley of $[Ca]_{sub}$. B. $V_m$ and $[Ca]_{sub}$ versus time when $[Ca]_{sub}$ is clamped at the lowest (red), middle (green), and highest (blue) levels during the plateau phase. C. $V_m$ and $[Ca]_{sub}$ versus time when $V_m$ is clamped at lowest (red) and highest (blue) levels in the plateau phase. D. $[Ca]_{sub}$ versus $V_m$ from simulations using the $V_m$ clamp protocol in which $V_m$ is switched from -80 mV to different clamped levels. Black dots: $I_{NCX}$ = 0. Red dots: $I_{Ca,L}$ = 0. The inset is a schematic plot of the feedback loop.

repeats until $[Ca]_{sub}$ accumulates to a high level which overly suppresses $I_{Ca,L}$ and repolarizes the AP. The effects of $[Ca]_{sub}$ on $I_{Ca,L}$ and thus $V_m$ can be seen in the $[Ca]_{sub}$ clamp simulation shown in Fig 5B, in which when $[Ca]_{sub}$ is clamped at a low level, $V_m$ fails to repolarize, but clamped at a high level, $V_m$ repolarizes normally.

## EADs driven by a large Ca transient (type IV)

If we increase $k_{max}$ (increase SR release) and/or reduce $J_{Caslmyo}$ (slow Ca diffusion from the submembrane space to cytoplasm) to increase Ca concentration in the submembrane space, another type of EADs occurs. This type of EADs occurs in the cyan strip in the phase diagram shown in Fig 6A. The characteristic feature of this type of EADs is that there is no oscillation in $[Ca]_{sub}$ but an EAD occurs in the AP (Fig 6B). The occurrence of the EAD is sensitive to $[Ca]_{sub}$ and if we reduce $[Ca]_{sub}$ (which is a clamped trace from the trace in Fig 6B with a 50% amplitude), no EAD occurs (Fig 6C). If we reduce either $I_{Ca,L}$ or $I_{NCX}$, the EAD amplitude is attenuated. Therefore, the mechanism of this type of EADs is as follows: Enhanced $[Ca]_{sub}$ increases $I_{NCX}$, which helps the reactivation of $I_{Ca,L}$ to reverse the $V_m$ decay, generating an EAD. This mechanism of EADs may be responsible for the late phase 3 EADs observed in

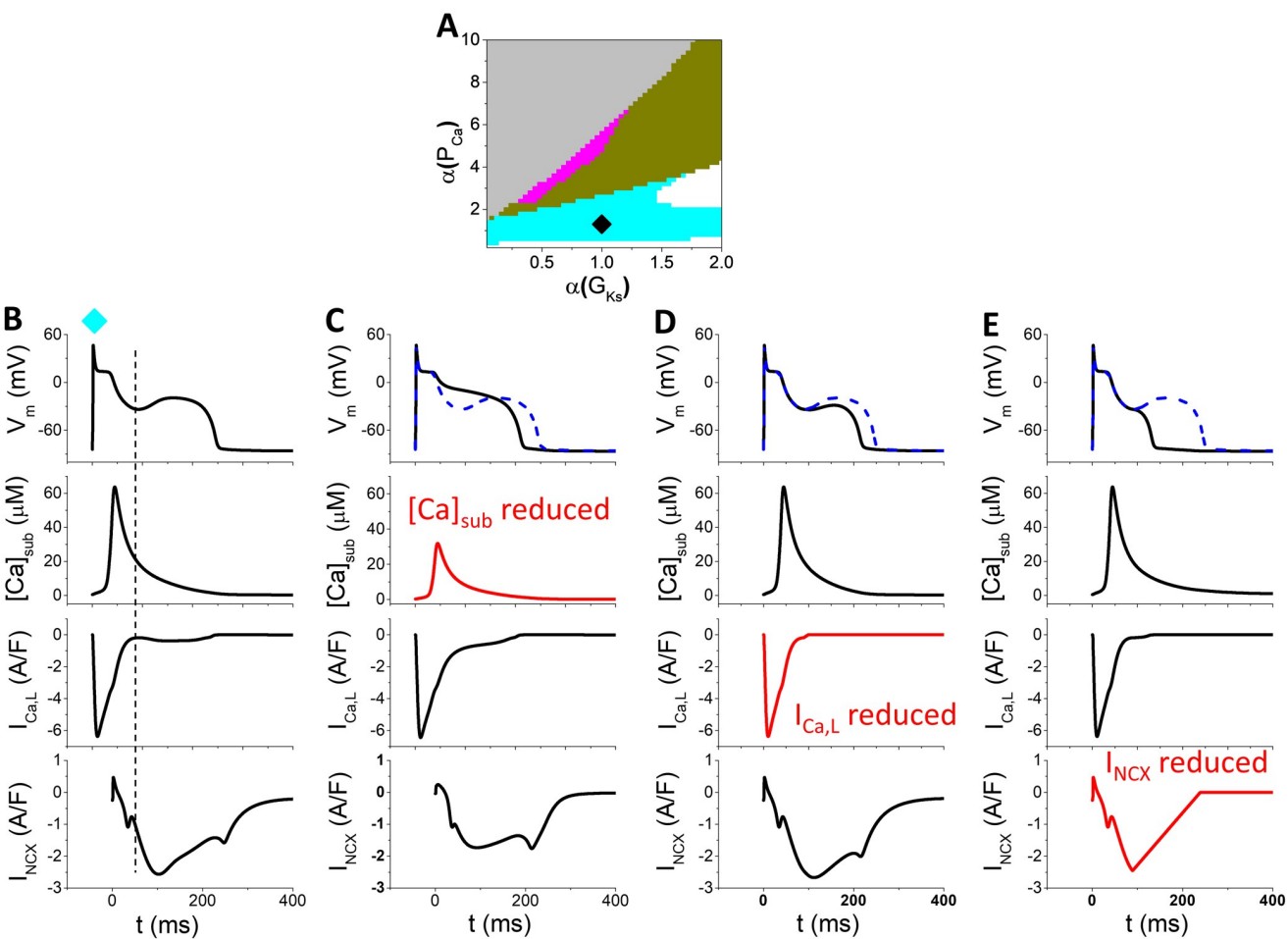

**Fig 6. EADs driven by a large Ca transient.** $\alpha(k_{max}) = 7$, $\alpha(J_{Caslmyo}) = 1$, and $\alpha(I_{NCX}) = 1.2$. A. Phase diagram showing different mechanisms of EAD on the $\alpha(P_{Ca}) - \alpha(G_{Ks})$ plane. The Ca transient driven EADs occur in the cyan region. Other colors are the same as indicated in the phase diagram in Fig 2A. B. $V_m$, $[Ca]_{sub}$, $I_{Ca,L}$, and $I_{NCX}$ versus time for a representative case $[\alpha(G_{Ks}) = 1$ and $\alpha(P_{Ca}) = 1$, diamond in A]. The vertical dashed line indicates the EAD takeoff moment. C. Same as B but $[Ca]_{sub}$ is clamped to 0.5 times of the trace in B (highlighted by red). The blue dashed $V_m$ trace in the $V_m$ panel is the original one for comparison. D. Same as B but $I_{Ca,L}$ is reduced linearly from the EAD takeoff moment. E. Same as B but $I_{NCX}$ is reduced linearly from the EAD takeoff moment.

experiments [41, 42] in which a large contraction or prolonged Ca transient causes phase 3 triggered activity or EADs.

## EAD properties of other ventricular AP models

We also investigate EAD properties using other ventricular AP models, including the phase II Luo and Rudy guinea pig model (LRd [43], later modified by Faber and Rudy [44]), Mahajan rabbit model [45] (later modified by Huang et al [46], $H_{UCLA}$), Ten Tuscher and Panfilov human model (TP04 [47]), O'Hara and Rudy human model (ORd [48]), and Grandi and Bers human model (GB [49]). These models belong to the second generation of cardiac AP models which incorporate detailed intracellular Ca cycling. Instead of plotting/scanning phase diagrams as for the WG model investigated above, we use a Monte Carlo approach by randomly assigning parameters between 0.1–10 times of their control values (see S1 Text). The types of

**Table 1. Mechanisms of EADs in different AP models.** For each model, a total of 10000 randomly selected parameter sets is simulated, and the number of parameter sets for which the APs exhibit EADs is indicated in the parentheses.

|  | type I | type II | type III | type IV |
|---|---|---|---|---|
| LRd (4030) | 100% | 0 | 0 | 0 |
| H$_{UCLA}$ (2329) | 99.87% | 0 | 0 | 0.13% |
| TP04 (2751) | 100% | 0 | 0 | 0 |
| ORd (2467) | 100% | 0 | 0 | 0 |
| GB (535) | 89.35% | 3.55% | 7.1% | 0 |
| WG (1113) | 13.44% | 38.2% | 7.45% | 40.91% |
| $f_{ss}$-flattened-GB (757) | 100% | 0 | 0 | 0 |
| $f_{ss}$-flattened-WG (1795) | 36.15% | 0 | 0 | 63.85% |

EADs are also identified by the analyses similar to that used for the WG model. The results are summarized in Table 1, and the typical APs are shown in supplemental Figs F-M in S1 Text. The WG model can exhibit 4 types of EADs, as already shown above. As expected, all models exhibit $V_m$-driven (type I) EADs. The H$_{UCLA}$ model can exhibit Ca transient driven (type IV) EADs with a very low probability, and the EAD amplitudes are extremely small (see Fig H in S1 Text). The GB model can exhibit both type II and type III EADs (see Figs L and M in S1 Text).

According to the results in Table 1, one can divide the models into two groups: I) the LRd model, the H$_{UCLA}$ model, the TP04 model and the ORd model; and II) the GB model and the WG model. In group I, the Ca system is stable, which cannot exhibit spontaneous Ca oscillations (it can passively follow $V_m$ oscillations as in type I EADs), whereas in group II, the Ca system can oscillate spontaneously. A major difference distinguishing the two groups is the steady-state inactivation curve ($f_{ss}$) for $I_{Ca,L}$. In group I, $f_{ss}$ is a typical decreasing sigmoidal function, but in group II, $f_{ss}$ first decreases but then increases with $V_m$, which is a non-monotonic function (see the different $f_{ss}$ curves in Fig B in S1 Text). This difference in $f_{ss}$ causes different responses of $I_{Ca,L}$ to $V_m$ and thus $[Ca]_{sub}$ to $V_m$ in the GB model and WG model from those in the group I (see Figs B and C in S1 Text), which allow spontaneous Ca oscillations to occur in group II. Moreover, the two models in group II also have similar Ca cycling models which are based on the SB model [38]. If we flatten the late elevated portion of $f_{ss}$ in the GB model and the WG model, type II and type III EADs cannot be observed (see Table 1).

## Discussion and conclusions

In this study, we carry out computer simulations of AP models with detailed intracellular Ca cycling to investigate the mechanisms of EADs. More specifically, we investigate the roles of Ca cycling and its coupling with $V_m$ in the genesis of EADs in cardiac myocytes. We first use the WG model and identify 4 mechanisms of EADs:

1. The first type of EADs is driven by oscillations originating from the $V_m$ system. The condition for this to occur is called RRR in which outward currents are reduced and/or inward currents are increased. RRR combined with proper $I_{Ca,L}$ kinetics provides a condition for a Hopf bifurcation to occur, leading to $V_m$ oscillations that manifest as EADs. The rise of $V_m$ in the EADs is mediated by the reactivation of $I_{Ca,L}$ and this type of EADs needs a high $I_{Ca,L}$ conductance (see Fig 2A). This is the type of EAD mechanism widely investigated in theoretical and simulation studies [20–22, 24], which may account for the mechanism for the majority of EADs observed in experimental studies [8, 9].

2. The second type of EADs is driven by oscillations originating from the Ca cycling system. This type of EADs tends to occur when SR release and $I_{NCX}$ are strong, but a larger than control $I_{Ca,L}$ is still needed (Figs 2A and 4A). Ca oscillations are promoted by strong SR release and the elevated Ca combined with a strong $I_{NCX}$ gives rise to $V_m$ depolarizations to manifest as EADs. Reactivation of $I_{Ca,L}$ is not required, however, a strong $I_{Ca,L}$ is still needed to hold $V_m$ at the plateau phase for Ca and $I_{NCX}$ to cause $V_m$ depolarizations. This type of EADs has been investigated in computer simulation studies [33–35] and experimental evidence has been shown in several experimental studies [29, 32]. However, a rigorous nonlinear dynamics analysis is still lacking to pinpoint the exact nonlinear dynamics/bifurcations for this mechanism of EADs.

3. The third type of EADs is driven by a feedback loop between $V_m$ and Ca mediated mainly by $I_{Ca,L}$. This type of EADs tends to occur when $I_{Ca,L}$ conductance is very high (Fig 2A). In this case, oscillations cannot be generated by either $V_m$ or Ca alone but are caused by their coupling. The exact mechanism of oscillations is unknown, and we postulate that the feedback between $V_m$ and Ca combined with a special steady-state inactivation curve of $I_{Ca,L}$ is the underlying driver for this type of EADs. To the best of our knowledge, there is no experimental evidence on this mechanism of EADs. Moreover, even if EADs of this mechanism occur in an experimental system, it will be difficult to distinguish them from the ones driven by the $V_m$ oscillations or the Ca oscillations.

4. The fourth type of EADs is driven by an enhanced Ca transient. This type of EADs occurs when the SR Ca release is very strong to result in an elevated and prolonged Ca transient. However, a stronger than normal $I_{Ca,L}$ or $I_{NCX}$ is not a requirement (Fig 6A) but both play an important role. Only a single EAD can be generated in an AP, and it is not an oscillation but a transient phenomenon. Experimental evidence for this EAD mechanism may be supported by the late phase 3 EADs observed in experimental studies [41, 42] in which the occurrence of a phase 3 EAD or triggered activity is companied by a large and prolonged contraction or Ca transient.

We also carry out simulations of 5 other AP models to assess the 4 mechanisms of EADs. We use a Monte Carlo approach to randomly select parameters in pre-assigned wide ranges. All models can exhibit the $V_m$ oscillation driven (type I) EADs but only the GB and WG models can exhibit the Ca oscillation driven (type II) and $V_m$-Ca feedback loop driven (type III) EADs. We identify that the difference in the steady-state inactivation curve of $I_{Ca,L}$ may be responsible for the different behaviors of the models since if we change the steady-state inactivation curve in WG (or GB) model to the same as in the other models, we are not able to observe both type II and type III EADs. Another difference is that the Ca cycling models in the WG model and the GB model are based from the one in the SB model which allows spontaneous Ca oscillations. The results of these simulations also raise a question on proper modeling of Ca cycling in the AP models, which needs to be addressed properly using advanced modeling approaches [50, 51].

Although we use computer simulations to scan parameters for phase diagrams and use a Monte Carlo approach to select parameters in wide ranges, rigorous nonlinear dynamics analyses are still needed to pinpoint the exact dynamics for EAD genesis. So far, such analyses have been mainly done for $V_m$ oscillation driven EADs. However, such analyses have been done mostly in simplified models with no Ca cycling. In the presence of Ca cycling, the models become too complex to be useful for such analyses, and simplified models are needed in future bifurcation analyses. In most of the simulations, we use a single set of initial condition and a single stimulus to elicit an AP. We then scan the parameters to assess the EAD

behaviors. However, the heart is under constant pacing, and a change in parameters will alter its equilibrium state, particularly those of the intracellular ions. Since the equilibrium state is heartrate dependent, and thus the EAD behaviors can also be heartrate dependent. Nevertheless, using different pacing protocols (Fig 3), we obtained similar phase diagrams to that using the single initial condition, indicating that although the EAD behaviors may be heartrate dependent, our simple pacing protocol can still reveal the underlying mechanisms of EADs.

Besides the insights gained on dynamical mechanisms of EAD genesis, our study may also provide insights into EAD genesis in pathological conditions. EADs are a hallmark of LQTS in which either inward currents are increased or outward currents are decreased, or both occur. Due to the main changes are membrane currents to prolong APD, the EAD mechanisms are mainly via $V_m$-driven oscillations (type I) [52]. However, experimental evidence of Ca-driven oscillations might also be responsible for EADs in LQTS [32]. EADs are widely observed in heart failure [53, 54]. In heart failure, remodeling occurs in both membrane ionic currents and Ca cycling [55], it is possible that types I-III EAD mechanisms can occur depending on specific heart failure conditions. EADs are also observed in catecholaminergic polymorphic ventricular tachycardia in which RyR mutations cause leaking RyRs to alter the Ca cycling behaviors and promote spontaneous Ca release [56, 57]. EADs under this diseased condition are likely caused by the Ca-driven oscillations (type II mechanism), however, this mechanism still requires an increase in $I_{Ca,L}$ (Fig 2A). In another diseased condition in which RyR mutations causes a reduction of RyR open probability and EADs [58]. Reduced RyR open probability increases the SR load which then results in a large Ca transient, and EADs can be caused by the large Ca transient as in the type IV EAD mechanism.

In conclusion, using computer simulations of AP models with detailed Ca cycling, we identify 4 mechanisms of EAD genesis arising from $V_m$ oscillations or Ca oscillations, or their coupling and feedback loops. The insights from this study provide theoretical bases for the EADs observed in experiments and deepen our understanding of the mechanisms of EADs and cardiac arrhythmogenesis.

## Supporting information

**S1 Text. Supplemental information.** The supplemental information contains the following information: i) Simulation results of bidirectional regulations of $V_m$ and submembrane Ca in different models; ii) the types of EADs; and iii) Control parameters and initial values of each model.
(PDF)

**S1 Code. Computational code of the WG model.**
(DOCX)

## Author Contributions

**Conceptualization:** Zhilin Qu, Xiaodong Huang.

**Data curation:** Zhilin Qu, Xiaodong Huang.

**Formal analysis:** Rui Wang, Xiaodong Huang.

**Funding acquisition:** Zhilin Qu, Xiaodong Huang.

**Investigation:** Rui Wang, Zhilin Qu, Xiaodong Huang.

**Methodology:** Rui Wang.

**Supervision:** Xiaodong Huang.

**Writing – original draft:** Zhilin Qu, Xiaodong Huang.

**Writing – review & editing:** Rui Wang, Zhilin Qu, Xiaodong Huang.

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
