## [Decision Letter · Decision Letter 0]

3 Jan 2024

Dear Dr. Huang,

Thank you very much for submitting your manuscript "Dissecting the roles of calcium cycling and its coupling with voltage in the genesis of early afterdepolarizations in cardiac myocyte models" for consideration at PLOS Computational Biology.

As with all papers reviewed by the journal, your manuscript was reviewed by members of the editorial board and by several independent reviewers. In light of the reviews (below this email), we would like to invite the resubmission of a significantly-revised version that takes into account the reviewers' comments.

We cannot make any decision about publication until we have seen the revised manuscript and your response to the reviewers' comments. Your revised manuscript is also likely to be sent to reviewers for further evaluation.

Sincerely,

Jeffrey J. Saucerman

Academic Editor

PLOS Computational Biology

Jason Haugh

Section Editor

PLOS Computational Biology

Reviewer's Responses to Questions

**Comments to the Authors:**

Reviewer #1: In the article entitled “Dissecting the roles of calcium cycling and its coupling with voltage in the genesis of early afterdepolarizations in cardiac myocyte model’’ by Wang et al, the authors categorize four different mechanisms for the development of EAD. The reviewer believes that this paper is well written; however, this paper contains problems with methodological rigor, and the reviewer has concerns about the results and its conclusions. Therefore, I do not think they are suitable for publication as is in PLoS Computational Biology.

The main concern of this manuscript is that the methodology adopted in the present study includes a certain kind of problem. It is a well-known fact that initial conditions (initial value sets of state variables) are extremely important in simulations (numerical integrations) in ordinary differential equation models. Depending on the parameter set, it is not surprising to observe bistable or tri-stable dynamics depending on their initial values (e.g. Tsumoto et al., Sci Reps, 2017). The results presented in this manuscript shown here have been obtained by a huge number of simulations though, no data are provided regarding the initial values at which the simulations were performed. The authors should provide initial values (initial conditions) for each simulation.

As a question, were these simulations performed with identical initial values (initial conditions) for each simulation? If so, how significant is it to classify the transient dynamics obtained by single stimulation under identical initial conditions? For example, could changing the initial conditions change dynamics classified as Vm-driven mechanisms to dynamics due to other EAD mechanisms? To dispel these doubts, authors should classify the mechanism of EAD development when the action potential response of cardiomyocyte models reaches a steady state after repeated application of multiple stimuli. The reviewer believes that the mechanisms of EAD at steady-state would be most reliable if they could be explained individually.

Minor.

1. The reviewer could not understand the schematic in Fig.1A. What does the outermost line mean? (Does it mean the cell membrane?) If so, what space does the SL space between the myoplasm and the plasma membrane mean?

2. Does JXN space mean the subspace in the cytoplasm where the t-tubule membrane and SR membrane (junctional SR membrane) are close together?

3. Does JCaslmyo mean Ca influx into the cell via L-type Ca channels?

4. To put a finer point on it, the reviewer can imagine that ‘ctrl’ means ‘control’, but it should be defined properly.

5. The meaning of the lines in the lower panel in Fig. 2C are reversed: the blue line is the highest and the red line is the lowest level.

6. In each phase diagram, what dynamics occur in the gray region? Please add an explanation.

Reviewer #2: In this study by Wang and colleagues, the authors investigate early afterdepolarizations (EADs) in cardiac myocyte models, considering different parameter regimes and highlighting 4 different mechanisms underlying EADs.

The study demonstrates both previously identified and new mechanisms for EAD generation, the respective parameter conditions needed for each mechanism, and also identifies the presence (or absence) of these mechanisms in several different well-established cardiac models.

The study is rigorous and the manuscript well written. My comments are primarily focused on data presentation and clarification of methods.

1. In Figure 2A, please clarify the model dynamics in the grey and white regions of the phase diagram (either in the text or figure caption). Presumably, these are repolarization failure and normal APs, but please clarify.

2. Methods: Was some automated analysis used to identify the mechanisms in the different phase diagrams and the Monte Carlo study (of other models), or were all simulations analyzed individually? Please clarify.

If automated, please describe the specific criteria used for classification in the Methods.

3. Figure 3B: Can the authors add another vertical line near the start of the oscillations? It is not obvious that the initial increase in Ca_sub leads the initial increase in Vm, which is a key aspect of this mechanism.

4. Table 1: Please clarify how many simulations were performed for each model using the Monte Carlo approach.

5. As a general comment, the Discussion and Conclusions section is quite brief. I appreciate the succinct summary of the 4 mechanisms, but I think it would be valuable for the authors to provide some additional physiological context to the study, in particular related to the feasibility of the different EAD mechanisms under different pathological conditions.

Minor: In Figure 2C, please confirm the 'lowest' and 'highest' labels in the caption. They appear to be switched.

Minor: Figure 3E: 'clamped' is misspelled in the figure titles

**Have the authors made all data and (if applicable) computational code underlying the findings in their manuscript fully available?**

Reviewer #1: **No: **While the availability of data is stated that “all relevant data are within the manuscript and its supporting information files,” there is no mention of the availability of codes.

Reviewer #2: **No: **No code is provided as a supplement or in a public repository. It would be appropriate to provide code for the models used in the study and EAD analysis classification (if used, see comments).

PLOS authors have the option to publish the peer review history of their article (what does this mean?). If published, this will include your full peer review and any attached files.

Reviewer #1: **Yes: **Kunichika Tsumoto

Reviewer #2: No
---

## [Decision Letter · Decision Letter 1]

19 Feb 2024

Dear Dr. Huang,

We are pleased to inform you that your manuscript 'Dissecting the roles of calcium cycling and its coupling with voltage in the genesis of early afterdepolarizations in cardiac myocyte models' has been provisionally accepted for publication in PLOS Computational Biology.

Best regards,

Jeffrey J. Saucerman

Academic Editor

PLOS Computational Biology

Jason Haugh

Section Editor

PLOS Computational Biology

Reviewer's Responses to Questions

**Comments to the Authors:**

Reviewer #1: The authors have adequately addressed the issues raised by the reviewer.

Reviewer #2: The authors have addressed all of my concerns.

**Have the authors made all data and (if applicable) computational code underlying the findings in their manuscript fully available?**

Reviewer #1: Yes

Reviewer #2: Yes

PLOS authors have the option to publish the peer review history of their article (what does this mean?). If published, this will include your full peer review and any attached files.

Reviewer #1: No

Reviewer #2: **Yes: **Seth H. Weinberg

---

## [Editor Report · Acceptance letter]

23 Feb 2024

PCOMPBIOL-D-23-01792R1 

Dissecting the roles of calcium cycling and its coupling with voltage in the genesis of early afterdepolarizations in cardiac myocyte models

Dear Dr Huang,

I am pleased to inform you that your manuscript has been formally accepted for publication in PLOS Computational Biology. Your manuscript is now with our production department and you will be notified of the publication date in due course.

With kind regards,

Zsofia Freund
